# Efficacy and safety of antifibrotic drugs for interstitial lung diseases other than IPF: A systematic review, meta-analysis and trial sequential analysis

**Mei Yang[1], Yuying Tan[1], Ting Yang[1], Dan Xu[2], Mei Chen[3]\*, Lei Chen [1]\***

1 Department of Pulmonary and Critical Care Medicine, West China Hospital, Sichuan University, Chengdu, Sichuan, China, 2 Laboratory of Pulmonary Diseases, West China Hospital, Sichuan University, Chengdu, Sichuan, China, 3 School of Medical and Life Sciences, Chengdu University of Traditional Chinese Medicine, Chengdu, Sichuan, China

☯ These authors contributed equally to this work.

\* lchens@126.com (LC); alice_chenmei@163.com (MC)

## Abstract

### Background

The therapeutic role of antifibrotic therapy has been well-established in idiopathic pulmonary fibrosis (IPF). However, its efficacy and safety for interstitial lung diseases (ILDs) other than IPF are not fully understood.

### Methods

We updated a systematic review with meta-analysis and trial sequential analysis (TSA) of randomized controlled trials and prospective studies on antifibrotic drug (nintedanib or pirfenidone) vs other intervention (placebo, no intervention or conventional treatment) in non-IPF ILDs. The primary outcomes were absolute change in forced vital capacity (FVC), all-cause mortality and serious adverse events (SAEs). The risk of bias was rated with the RoB2 tool and certainty of evidence was assessed by the GRADE approach.

### Results

17 studies with 1908 patients were included. For the primary outcomes, pooled analyses of four trials with low risk of bias showed that antifibrotic drugs significantly ameliorated FVC decline (mean difference 86.21; 95% CI 49.38 to 123.03; I² = 64%; TSA-adjusted CI 40.86 to 131.56). Based on five trials with low risk of bias, no difference was observed in all-cause mortality (RR 0.87; 95% CI 0.53 to 1.43; I² = 0%; TSA-adjusted CI 0.12 to 6.53) and SAEs (RR 0.97; 95% CI 0.83 to 1.13; I² = 0%; TSA-adjusted CI 0.74 to 1.28) between groups. However, based on two studies with 324 patients, benefit of antifibrotic drugs in FVC was not shown in the subgroup taking mycophenolate (mean difference 17.08; 95% CI -56.22 to 90.37), which also had higher risk of SAEs (RR 1.71; 95% CI 1.09 to 2.70), although both were contested by TSA.

**Data Availability Statement:** The data underlying this study are available in the manuscript and its online supplementary material.

**Funding:** The author(s) received no specific funding for this work.

**Competing interests:** The authors have declared that no competing interests exist.

## Conclusion

Our study suggests that antifibrotic drugs are beneficial for patients with non-IPF ILDs in slowing disease progression, whereas may not correlate to all-cause mortality and SAEs. However, for patients taking mycophenolate, antifibrotic drugs may do more harm than good. More investigations are warranted to validate current findings.

## Introduction

Interstitial lung diseases (ILDs) other than idiopathic pulmonary fibrosis (IPF) have imposed growing disease burden [1,2]. The non-IPF phenotype encompasses a wide range of rare ILDs, such as autoimmune disease-related ILD (AID-ILD) and exposure-related ILD among other groups [3]. Inflammation and fibrosis are considered as the primary pathophysiological features of ILDs [3,4]. Similar to IPF, non-IPF ILDs may develop progressive pulmonary fibrosis, which has contributed substantially to the high disability and mortality [2,5]. As reported, nearly 15% of new referrals with non-IPF ILDs developed progressive fibrosis though receiving standard therapy [2]. Treatment strategies for non-IPF ILDs are now limited, which are mainly based on glucocorticoids and sometimes immunosuppressive therapy, whereas the efficacy and safety remain controversial [6,7].

Nintedanib and pirfenidone are antifibrotic drugs widely used in IPF. The antifibrotic and anti-inflammatory properties of both drugs have been shown to slow fibrosis progression in patients with IPF [8]. Of note, emerging evidence has indicated similar pathogenic mechanisms across a variety of ILDs, especially fibrogenesis [3,9]. Clinically, patients with a non-IPF phenotype also resemble IPF in radiological-histopathological pattern and disease course [3]. Those make it plausible for the use of nintedanib and pirfenidone in non-IPF ILDs.

Current evidence concerning antifibrotic drugs in the non-IPF phenotype has been limited. The only two randomized controlled trials (RCTs) with large sample sizes, SENSCIS [10,11] and INBUILD trials [12–15], assessed the efficacy of nintedanib in patients with systemic sclerosis-associated ILD (SSc-ILD) and progressive fibrosing ILD, and revealed the benefit of nintedanib in improving annual decline rate of forced vital capacity (FVC). Whereafter, several studies also suggested the potential of pirfenidone in slowing disease progression, with acceptable safety [16–18]. However, evidence-based meta-analyses (a total of four studies included) indicated uncertain results due to low quality of evidence [19,20]. Similar low certainty was also highlighted in the ATS and ATS/ERS/JRS/ALAT guidelines [5,7].

Recently there is a rise in relevant RCTs, further providing data on non-IPF ILDs. Therefore, we conduct this systematic review and meta-analysis, aiming to summarize and update the evidence regarding efficacy and safety of antifibrotic drugs, nintedanib and pirfenidone, in patients with ILDs other than IPF.

## Methods

This study was conducted and reported according to the Cochrane Handbook [21] and the Preferred Reporting Items for Systematic Reviews and Meta-Analyses guidelines (S1 Checklist) [22]. The protocol was registered in the PROSPERO repository (CRD42024542367). More details were presented in the S1 File.

## Eligibility criteria

1. Study design: RCTs and prospective studies.

2. Participants: adult patients with non-IPF ILDs, including AID-ILD, exposure-related ILD and sarcoidosis etc. [3], with no restriction on clinical phenotype, histopathology or extent of fibrosis.

3. Intervention: nintedanib or pirfenidone, independent of the dose and duration.

4. Comparison: placebo, conventional treatment recommended by the guidelines, internationally recognized treatment, or no intervention.

5. Outcomes: primary outcomes were absolute change in FVC (ml) from baseline to study endpoint, all-cause mortality and serious adverse events (SAEs). Secondary outcomes included absolute changes in FVC% predicted, diffusing capacity of the lung for carbon monoxide (DLCO) % predicted, six-minute walk distance (6MWD) and St. George's Respiratory Questionnaire (SGRQ), from baseline to study endpoint; annual rates of decline in FVC and FVC% predicted, acute exacerbation of ILD, adverse events (AEs) and AEs leading to treatment discontinuation. The exploratory outcome was respiratory-related mortality, predefined as the lower respiratory disease or pulmonary vascular disorder being assessed as a primary, underlying, or contributing cause of death. All the outcomes were assessed at the latest time point within 6 to 12 months, unless otherwise stated.

## Search strategy and study selection

Four electronic databases (PubMed, Ovid EMBASE, Cochrane Library and ClinicalTrials.gov) were searched using medical subject headings or key words. For the purpose of rapid review, the language was restricted to English [23]. No restriction was applied for the publication status or year. The last search was conducted on May 20, 2024. The full search strategy is available in the S1 File.

According to the inclusion criteria, two investigators (MY and YT) independently performed systematical search, screened titles and abstracts of all retrieved publications to exclude duplicate or irrelevant records. For articles requiring further assessment, full-text reviews were carried out. The references of included articles and relevant reviews were also screened to identify additional eligible studies. Disagreement was addressed by discussion between the two reviewers or with the help of the third investigator (TY).

## Data extraction and risk-of-bias assessment

Two investigators (MY and YT) independently extracted data from selected studies using a standardized collection form. The following information were extracted: study characteristics (author, year of publication, country, design, sample size and duration), patient characteristics (demographics, inclusion and exclusion criteria, clinical, radiological and if available, pathological manifestations), ILD subtype, intervention (nintedanib or pirfenidone, dose, frequency, duration) and control, predefined outcome measures. If the mentioned data was not available, corresponding authors were contacted for more information.

Risk of bias for each study was independently evaluated by two investigators (TY and DX). The Cochrane Collaboration tool, RoB2, was applied to assess the quality of RCT based on five domains, comprising randomization process, deviations from intended interventions, missing outcome data, measurement of the outcome and selection of the reported result. Trial was rated as low risk when all the five domains were at low risk of bias. Trial was rated as some

concerns or high risk if any domain was at some concerns or high risk. For observational studies, the Newcastle-Ottawa Scale was adopted [24], with a total score of 9. Considering the potential risk of bias for a non-RCT design, study with a total score of >5 was considered some concerns, whereas a score of ≤ 5 was considered high risk of bias. Any discrepancy regarding data extraction and quality assessment was reconciled by a third author (MC).

## Data analysis

We planned to analyze the predefined outcome measures in all patients with non-IPF ILDs and those with a progressive fibrosing phenotype, respectively. The meta-analysis was performed using Review Manager version 5.4 (Cochrane Collaboration). For dichotomous outcome, the pooled risk ratio (RR) with 95% CI was used as the effect estimate, and for continuous outcome, weighted mean difference (MD) was applied. The Mantel-Haenszel method was applied to calculate the RRs and Inverse Variance method was used for the MDs. We applied both fixed and random effects models to test robustness of estimates, and reported the most conservative estimate based on the highest p value [25]. Statistical heterogeneity across studies was tested using inconsistency ($I^2$) and diversity ($D^2$) statistics. Publication bias was evaluated by funnel plot and Harbord test when 10 or more studies were included in a meta-analysis, using Stata version 16 (StataCorp), and p <0.05 was considered statistically significant.

## Trial sequential analysis

Considering that meta-analyses with small sample sizes and cumulative meta-analysis (repeated significance testing) could result in type I errors, we conducted trial sequential analysis (TSA) to evaluate the risk of random errors and conclusiveness of results. TSA could estimate the required information size (RIS) needed to reach credible results, with an adjusted threshold for statistical significance. We quantified the trial sequential monitoring boundaries using a β of 10% (power of 90%) and a family-wise error rate of 5%, thus the α for primary, secondary and exploratory outcomes were 2.5%, 0.71%, and 5% respectively [25,26]. For dichotomous outcomes, we used continuity correction of 0.5 for zero event trial, and calculated the RIS using a relative risk reduction (RRR) based on trials with low bias (for primary outcomes also used an anticipated RRR of 20%). For continuous outcomes, we calculated the MD (Empirical) and variance (Empirical). The TSA was conducted for trials with low risk of bias only, using the TSA program version 0.9.5.10 (Copenhagen Trial Unit).

## Subgroup and sensitivity analyses

We planned to conduct subgroup analyses in all participants and those with a progressive fibrosing phenotype respectively, regarding: 1) risk of bias (low risk vs some concerns or high risk), 2) antifibrotic drugs (pirfenidone vs nintedanib), 3) follow-up time (<12 months vs ≥ 12 months), 4) ILD subtype, 5) HRCT pattern (usual interstitial pneumonia [UIP] vs non-UIP), 6) taking vs not taking mycophenolate at baseline. We used Chi-squared test to evaluate the statistical heterogeneity across subgroups with p<0.10 indicating significant. For sensitivity analysis, we performed pooled analyses of studies with low risk of bias and applied different statistical models (fixed and random effects) for all outcomes.

## Certainty of evidence assessment

We applied the Grading of Recommendations, Assessment, Development, and Evaluations (GRADE) tool to assess certainty of evidence, based on studies with low risk of bias only [27]. The certainty of evidence was rated as high, moderate, low, or very low.

## Results

The database searching yield 7000 records. After removing the duplicates, 4646 records were screened based on titles and abstracts, among which 4241 were excluded. The remained 405 full-text articles were assessed according to the eligibility criteria and 388 of them were further excluded. Finally, a total of 17 studies [10–18,28–35] with 1908 patients were included (Fig 1 and S1 Table).

### Characteristics of included studies

As shown in Tables 1 and S2, the 17 studies comprised 10 RCTs [10,12,16–18,29–31,34,35], five post hoc analyses of RCTs [11,13–15,32] and two prospective studies [28,33]. These studies

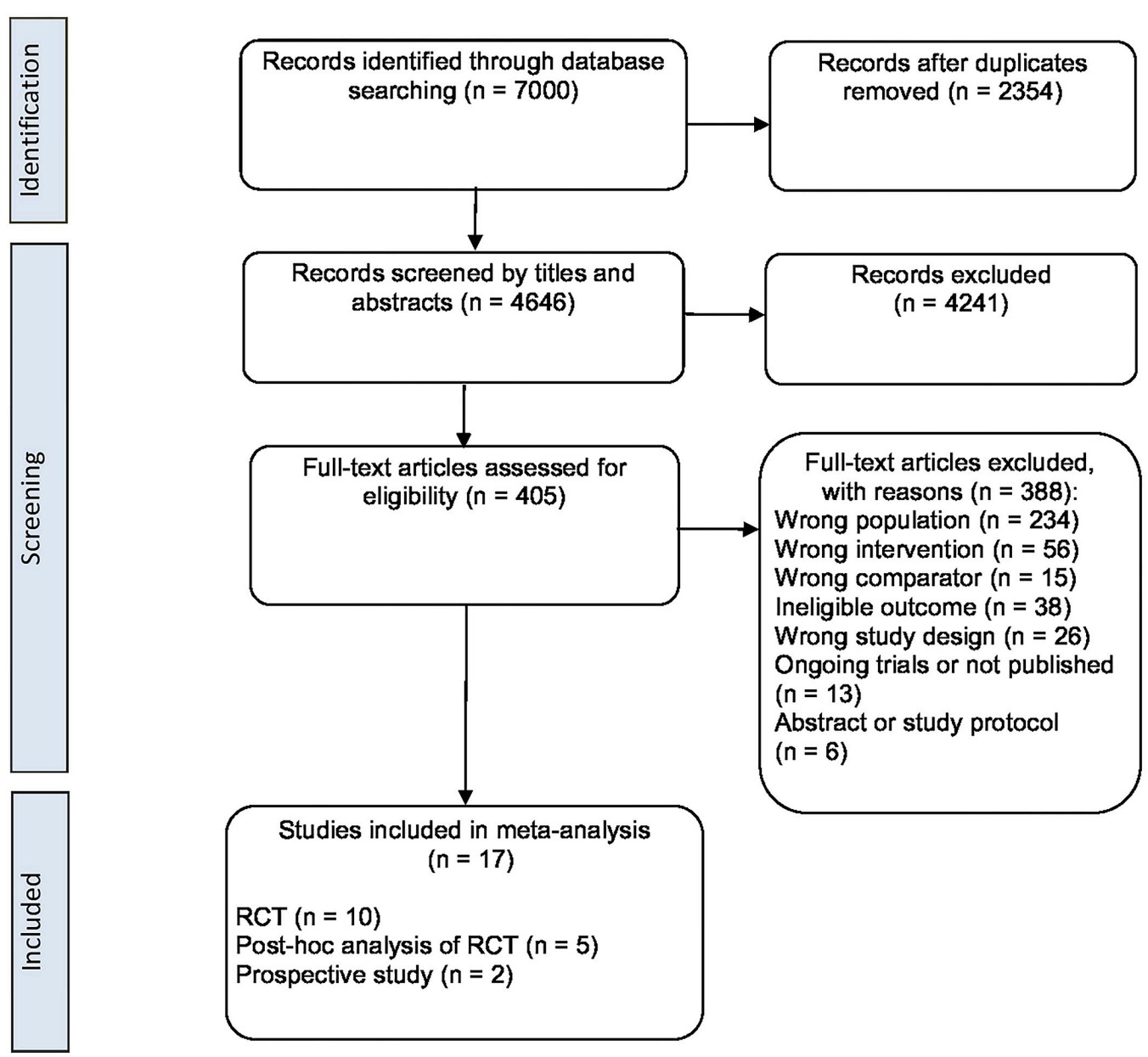

**Fig 1. Preferred Reporting Items for Systematic Reviews and Meta-analyses flow chart showing study screening and inclusion.**

**Table 1. Characteristics of included studies.**

| Study | Location | Recruitment Period | Study design | No. of Patients (Intervention / Control) | Age*, yr; Male sex, % | ILD subtypes; UIP pattern, % | Intervention vs Control; Follow-up time | Outcomes |
|---|---|---|---|---|---|---|---|---|
| Distler et al [22], 2019 (SENSCIS trial) | 32 countries | November 2015—October 2017 | Multicenter RCT | 576 (288 / 288) | 54.0 (12.2); 24.8% | SSc-ILD; NR | Nintedanib 150mg twice daily vs placebo; 52w | Annual rates of decline in FVC and FVC% predicted, change in FVC, DLCO% predicted, the modified Rodnan skin score, SGRQ and net digital ulcer burden; adverse event, time to death from any cause |
| Flaherty et al [23], 2019 (INBUILD trial) | 15 countries | February 2017—April 2018 | Multicenter RCT | 663 (332 / 331) | 65.8 (9.8); 53.7% | Progressive fibrosing ILDs other than IPF; 62.1% | Nintedanib 150mg twice daily vs placebo; 52w | Annual rate of decline in FVC, change in total score of K-BILD, acute exacerbation of ILD, death, adverse events |
| Acharya et al [24], 2020 | India | July 2017 -December 2018 | Single-center RCT | 34 (17 / 17) | 41.0 (20.0–63.0); 8.8% | SSc-ILD; 32.3% | Pirfenidone 200mg three times/day and increased by 600 mg/day every week to a target dose of 2,400mg/day vs placebo; 6m | The proportion of patients with improvement or stabilization in FVC, change in FVC, 6MWD, serum levels of TGF-β and TNF-α; adverse events |
| Maher et al [10], 2020 (trial NCT03099187) | 14 countries | May 2017 - June 2018 | Multicenter RCT | 253 (127 / 126) | 70.0 (61.0–76.0) vs 69.0 (63.0–74.0); 54.9% | Unclassifiable progressive fibrosing ILDs; NR | Pirfenidone 2,403mg daily vs placebo; 24w | Change in FVC, FVC% predicted, DLCO% predicted and 6MWD; proportion of patients with decline in FVC⩾5% or 10% predicted, all-cause and respiratory hospital admission, the incidence of, and time to first acute exacerbation, progression-free survival, adverse events |
| Mateos-Toledo et al [25], 2020 | Mexico | July 2015—March 2018 | Single-centre RCT | 22 (13 / 9) | 55.0 (7.0) vs 57.0 (9.0); 27.3% | cHP; NR | Pirfenidone 900mg twice a day plus conventional treatment vs conventional treatment only; 12m | Change in FVC% predicted, FVC, DLCO% predicted, oxygen saturation, 6MWD, SGRQ and HRCT score; adverse events |
| Behr et al [11], 2021 | Germany | April 2016—October 2018 | Multicenter RCT | 127 (64 / 63) | 63·2 (10·6) vs 63·5 (9·1); 59.1% | Progressive fibrosing ILDs other than IPF; NR | Pirfenidone 267mg three times/day (first week), 534mg three times/day (second week), 801mg three times/day (thereafter) vs placebo; 48w | Change in FVC% predicted, DLCO% predicted, 6MWD; progression-free survival, categorical assessment of relative change in FVC% of less than 5%, 5% to less than 10% and at least 10% predicted, SGRQ, time to clinical deterioration, adverse events |

*(Continued)*

**Table 1.** (Continued)

| Study | Location | Recruitment Period | Study design | No. of Patients (Intervention / Control) | Age*, yr; Male sex, % | ILD subtypes; UIP pattern, % | Intervention vs Control; Follow-up time | Outcomes |
|---|---|---|---|---|---|---|---|---|
| Shebl *et al* [28], 2021 | Egypt | December 2019—June 2020 | Single-centre RCT | 40 (20 / 20) | 48.7 (8.6) vs 44.6 (7.5); 67.5% | Progressive cHP; NR | Pirfenidone plus conventional treatment vs conventional treatment only; 6m | Change in FVC, 6MWD, partial pressure of oxygen in arterial blood and SGRQ; pulmonary artery systolic pressure with an echocardiogram radiological changes in HRCT chest |
| Fernández Pérez *et al* [33], 2023 | United States | June 2017 - April 2020 | Single-centre RCT | 40 (27 / 13) | 67.4 (6.5) vs 66.5 (3.6); 42.5% | Fibrotic HP; NR | Pirfenidone 267mg three times/day (two week), 534mg three times/day (two week), 801mg three times/day (thereafter) vs placebo; 52w | Change in FVC% predicted, FVC slope, DLCO% predicted and SGRQ; progression-free survival, all-cause hospitalisation, proportion of patients with progression in fibrosis on HRCT scans, adverse events |
| Rimner *et al* [34], 2023 | United States | October 2015—February 2020 | Multicenter RCT | 30 (18/12) | 72.0 (47.0–86.0); 27.0% | RP; NR | Nintedanib 150mg twice a day for 12 weeks plus standard 8-week prednisone vs placebo plus standard 8-week prednisone; 13m | Proportion of patients free from acute exacerbation, total number of exacerbation, pulmonary function tests, SGRQ, adverse events |
| Solomon *et al* [12], 2023 | 34 centers in four countries (Australia, Canada, the UK, the USA) | May 2017—March 2020 | Multicenter RCT | 123 (63 / 60) | 66·0 (61·0–74·0) vs 69·5 (63·5–74·5); 62.6% | RA-ILD; 65.9% | Pirfenidone, 267mg three times/day (first week), 534mg three times/day (second week), 801mg three times/day (thereafter) vs placebo; 52w | The incidence of decline in FVC ≥10% predicted or death, change in FVC, FVC% and dyspnoea; the frequency of categorical decline in FVC% or DLCO%, proportion of patients with a decline in FVC ≥10% predicted, adverse events |
| Flaherty *et al* [29], 2022 | 15 countries | February 2017 - April 2018 | Post-hoc analysis of the INBUILD trial | 663 (332 / 331) | 65.8 (9.8); 53.7% | Progressive fibrosing ILDs other than IPF; 62.1% | Nintedanib 150mg twice daily vs placebo; ≥19m | Declines in FVC ≥5% and 10% predicted, death, death, acute exacerbation of ILD, adverse events |
| Kreuter *et al* [30], 2022 | 14 countries | May 2017 - June 2018 | Subgroup analysis of trial NCT03099187 | 253 (127 / 126) | 70.0 (61.0–76.0) vs 69.0 (63.0–74.0); 54.9% | Unclassifiable progressive fibrosing ILDs; NR | Pirfenidone 2,403mg daily vs placebo; 24w | Change in FVC, DLCO% predicted and 6MWD; adverse events |
| Matteson *et al* [31], 2022 | 15 countries | February 2017—April 2018 | Subgroup analysis of the INBUILD trial | 170 (82 / 88) | 64.3 (10.6); 47.1% | AID-ILD; 74.7% | Nintedanib 150mg twice daily vs placebo; 52w | Rate of decline in FVC, change in FVC, FVC% and K-BILD; proportions of subjects with declines or increases in FVC >0% to ≤5%, >5% to ≤10%, >10% to ≤15%, and >15% predicted; death, adverse events |

(*Continued*)

**Table 1.** (Continued)

| Study | Location | Recruitment Period | Study design | No. of Patients (Intervention / Control) | Age*, yr; Male sex, % | ILD subtypes; UIP pattern, % | Intervention vs Control; Follow-up time | Outcomes |
|---|---|---|---|---|---|---|---|---|
| Wells et al [26], 2020 | 15 countries | February 2017—April 2018 | Subgroup analysis of the INBUILD trial | 663 (332 / 331) | 65.8 (9.8); 53.7% | Progressive fibrosing ILDs other than IPF; 62.1% | Nintedanib 150mg twice daily vs placebo; 52w | Annual rate of decline in FVC, adverse events |
| Highland et al [27], 2021 | 32 countries | November 2015—October 2017 | Subgroup of the SENSCIS trial | 576 (288 / 288) | 54.0 (12.2); 24.8% | SSc-ILD; NR | Nintedanib 150mg twice daily vs placebo; 52w | Annual rate of decline in FVC and FVC%, change in FVC and SGRQ, the proportion of patients with absolute decreases in FVC ≥5% and 10% predicted, adverse events |
| Li et al [21], 2016 | China | June 2014—November 2015 | Single-center, prospective interventional study | 57 (30 / 27) | 46.3 (11.3) vs 51.8 (7.8); 38.6% | Progressive CADM; NR | Pirfenidone 200mg three times/day and increased to 600mg three times/day over 2 weeks plus conventional treatment vs conventional treatment; 12m | All-cause mortality, adverse events |
| Wang et al [32], 2022 | China | August 2019—May 2021 | Single-center, Prospective cohort study | 136 (64 / 72) | ≥ 18; 16.2% | CTD-ILD; 15.3% | Pirfenidone 300mg/day and increased to the maximum tolerable dosage or 1,800mg/day plus conventional treatment vs conventional treatment; 24w | Change in FVC% predicted, adverse events |

*Data are presented as mean (SD) or median (interquartile range), for total population or intervention vs control.

Abbreviations: AID-ILD, autoimmune disease associated interstitial lung disease; CADM, clinically amyopathic dermatomyositis; cHP, chronic hypersensitivity pneumonitis; CTD-ILD, connective tissue disease-associated ILD; DLCO, diffusion capacity of the lung for carbon monoxide; FVC, forced vital capacity; IPF, idiopathic pulmonary fibrosis; K-BILD, the King's Brief Interstitial Lung Disease; NR, not reported; RA-ILD, rheumatoid arthritis-associated ILD; RCT, randomized controlled trial; RP, radiation pneumonitis; SGRQ, St George's Respiratory Questionnaire; 6MWD, six-minute walk distance; SSc-ILD, systemic sclerosis-associated ILD; UIP, usual interstitial pneumonia.

were published from 2016 to 2023, covering patients from the Africa, Americas, Asia, Europe and Oceania. 11 studies were multicenter [10–18,32,35] and six studies [28–31,33,34] were single-centre. The sample size ranged from 22 to 663 participants across studies. Four studies reported patient population of the INBUILD trial [12–15] and two studies reported patients of the SENSCIS trial [10,11]. Each patient population was analyzed only once in pooled analyses.

Pirfenidone was used in 10 studies [16–18,28–34] and nintedanib was applied in seven studies [10–15,35]. The duration of treatment varied from 6 to 19 months. For ILD subtypes, eight studies reported AID-ILDs, including ILDs associated with SSc [10,11,15,29,33], inflammatory myopathy [28,33], rheumatoid arthritis [15,18,33] etc. Four studies reported exposure-related ILDs, including hypersensitivity pneumonitis [30,31,34] and radiation pneumonitis [35]. Two trials reported unclassifiable ILD [16,32]. Patients with an UIP-pattern on imaging was reported in five studies [12,14,15,18,29],

Progressive fibrosing ILDs were reported in 11 studies [12–18,28,31,32,34]. The key inclusion criteria of patients and definitions of progressive fibrosis across studies were presented in the S1 File.

## Risk of bias

Risk of bias for all outcomes were presented in S1 File. For all included RCTs, some concerns or high risk was judged because of bias in missing outcome data and unbalanced baseline features between antifibrotic and control groups.

## Primary outcomes

**Absolute change in FVC (ml).** Seven RCTs with 1208 patients reported absolute change in FVC, five studies [16–18,30,31] for pirfenidone and two studies [10,15] for nintedanib. Among them, four RCTs [10,15,16,31] were rated as low risk of bias. Meta-analyses of the four RCTs suggested that antifibrotic drugs significantly improved the decline in FVC, with a MD of 86.21 ml between antifibrotic and control groups (95% CI 49.38 to 123.03; $I^2 = 64\%$; n = 999) (Fig 2A). This was consistent with pooled analyses of all the seven trials (MD 85.40; 95% CI 61.11 to 109.68; $I^2 = 34\%$) and five trials on patients with progressive fibrosis (MD 99.98; 95% CI 97.94 to 102.02; $I^2 = 0\%$; n = 616). TSA for trials with low risk showed that the cumulative z curve crossed the TSA monitoring boundary for benefit, with more than the RIS of 651 patients accrued (TSA-adjusted CI 40.86 to 131.56) (Fig 2B). The certainty of evidence was moderate (Table 2).

Consistent results were shown in subgroup analyses, except for the subgroup taking mycophenolate at baseline (MD 17.08; 95% CI -56.22 to 90.37; $I^2 = 24\%$; two studies with low risk of bias; n = 324). TSA for this subgroup showed uncertain result, with only 3.7% of the RIS of 8781 patients accrued.

**All-cause mortality.** 10 studies with 1920 patients reported all-cause mortality, seven for pirfenidone [16–18,28–30,34] and three for nintedanib [10,12,35]. Across these studies, all-cause mortality ranged from 0 to 51.9% in control group, with 0–51.9% in AID-ILDs [10,15,18,28,29], 0–7.7% in hypersensitivity pneumonitis [30,34] and 0.8–51.9% in progressive fibrosing ILDs [12,16,17]. Meta-analyses of five RCTs [10,12,16,17,29] with low risk of bias revealed that antifibrotic drugs were not associated with all-cause mortality (RR 0.87; 95% CI 0.53 to 1.43; $I^2 = 0\%$; n = 1650), consistent with analyses of all the ten studies (RR 0.83; 95% CI 0.57 to 1.20; $I^2 = 0\%$) (Fig 3A) and six studies [12,16–18,28,34] on progressive fibrosing ILDs (RR 0.71; 95% CI 0.47 to 1.09; $I^2 = 0\%$; n = 1261). TSA indicated the cumulative z curve neither crossed the conventional nor the TSA boundary for benefit, harm or futility, with 8.2% of the RIS accrued (TSA-adjusted CI 0.12 to 6.53) (Fig 3B). The certainty of evidence was moderate (Table 2). The 10 studies reporting all-cause mortality were analyzed for publication bias. As results, the funnel plot and Harbord test (p = 0.93) did not indicate significant publication bias.

**Serious adverse events.** Nine studies with 1938 patients reported SAEs, six for pirfenidone [16–18,29,30,33] and three for nintedanib [10,12,35], in which SAEs were defined according the preferred terms in the Medical Dictionary for Regulatory Activities. Meta-analyses of five trials [10,12,16,17,29] with low risk of bias suggested antifibrotic drugs did not markedly increase the risk of SAEs (RR 0.97; 95% CI 0.83 to 1.13; $I^2 = 0\%$; n = 1650), consistent with analyses of all the nine studies (RR 0.98; 95% CI 0.84 to 1.14; $I^2 = 0\%$) (Fig 4A) and four studies [12,16–18] on progressive fibrosing phenotype (RR 0.92; 95% CI 0.77 to 1.09; $I^2 = 0\%$; n = 1163). TSA showed the cumulative z curve neither crossed the conventional nor the TSA boundary for benefit, harm or futility, with 47.4% of the RIS accrued (TSA-adjusted CI 0.74 to

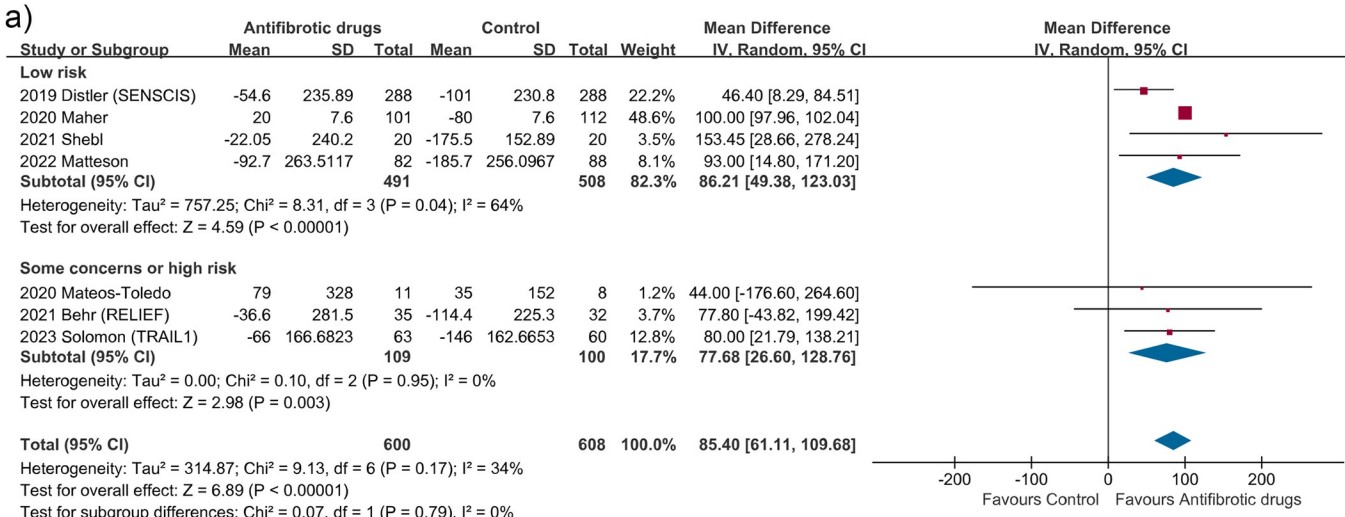

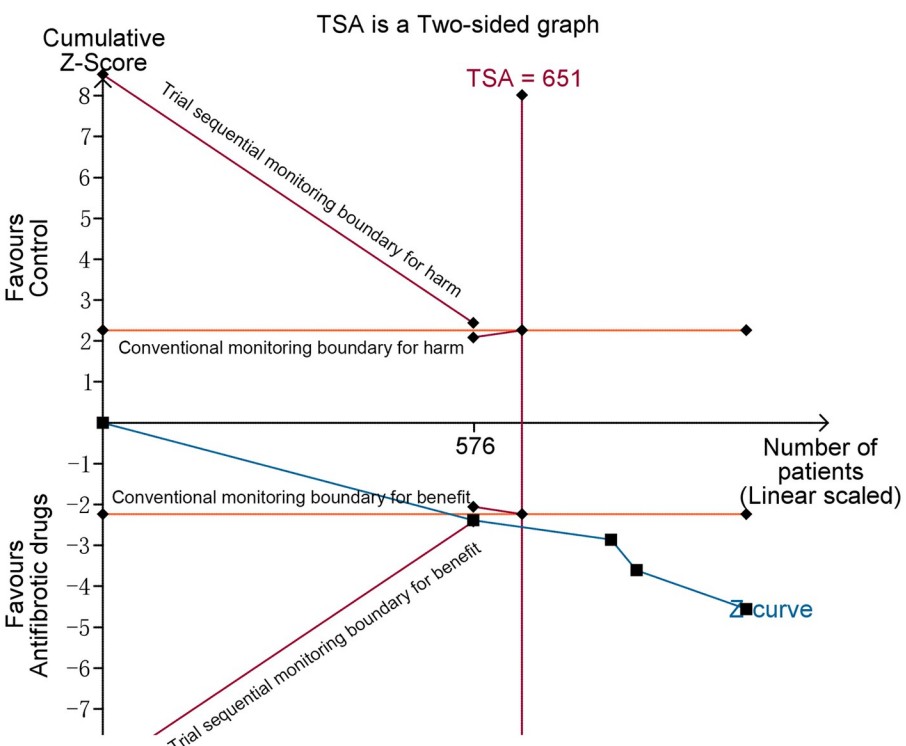

**Fig 2. Meta-analysis and trial sequential analysis (TSA) for absolute change in forced vital capacity (ml).** (a) Meta-analysis. (b) TSA for trials with low risk of bias. The required information size (RIS) was calculated based on mean difference (Empirical), variance (Empirical), α of 2.5% and β of 10%. The blue cumulative z curve crossed the TSA boundary for benefit, with more than the RIS of 651 patients accrued. Thus the TSA is conclusive, with the TSA-adjusted CI of 40.86 to 131.56 (random effects model) and a diversity $D^2$ of 100%.

1.28) (Fig 4B). The certainty of evidence was moderate (Table 2). Besides, 10 studies [10,12,16–18,29,30,33–35] reported fatal AEs and pooled analyses of those with low risk of bias also showed no statistical significance (RR 0.66; 95% CI 0.37 to 1.19; five studies

**Table 2. GRADE evaluation of the evidence in patients with ILDs other than IPF.**

| No. of Studies | Risk of bias | Inconsistency | Indirectness | Imprecision | Other considerations | Antifibrotic drugs | Placebo | RR (95% CI) | Absolute (95% CI) | Certainty | Importance |
|---|---|---|---|---|---|---|---|---|---|---|---|
| | | **Certainty Assessment** | | | | **No. of Patients** | | **Effect** | | | |
| **Absolute change in FVC (ml)** | | | | | | | | | | | |
| 4 | Not serious | Serious* | Not serious | Not serious | None | 491 | 508 | . . . | MD 86.21 higher (49.38 higher to 123.03 higher) | Moderate | Critical |
| **All-cause mortality** | | | | | | | | | | | |
| 5 | Not serious | Not serious | Not serious | Serious† | None | 28/827 (3.4%) | 32/823 (3.9%) | 0.87 (0.53-1.43) | 5 fewer per 1000 (from 18 fewer to 16 more) | Moderate | Critical |
| **SAEs** | | | | | | | | | | | |
| 5 | Not serious | Not serious | Not serious | Serious† | None | 221/827 (26.7%) | 227/823 (27.6%) | 0.95 (0.82–1.11) | 12 fewer per 1000 (from 43 fewer to 25 more) | Moderate | Critical |
| **Fatal AEs** | | | | | | | | | | | |
| 5 | Not serious | Not serious | Not serious | Serious† | None | 18/827 (2.2%) | 27/823 (3.3%) | 0.69 (0.38–1.26) | 10 fewer per 1000 (from 20 fewer to 8 more) | Moderate | Important |
| **Absolute change in FVC% predicted** | | | | | | | | | | | |
| 3 | Not serious | Serious‡ | Not serious | Not serious | None | 203 | 220 | . . . | MD 3.38 higher (1.24 higher to 5.53 higher) | Moderate | Important |
| **Absolute decline in FVC ≥ 10% predicted** | | | | | | | | | | | |
| 4 | Not serious | Not serious | Not serious | Not serious | None | 153/763 (20.1%) | 222/762 (29.1%) | 0.69 (0.58–0.81) | 80 fewer per 1000 (from 110 fewer to 48 fewer) | High | Important |
| **Annual rate of decline in FVC (ml/yr)** | | | | | | | | | | | |
| 2 | Not serious | Very serious¶ | Not serious | Serious† | None | 620 | 619 | . . . | MD 73.39 higher (8.62 higher to 138.15 higher) | Very low | Important |
| **Annual rate of decline in FVC% predicted** | | | | | | | | | | | |
| 1 | Not serious | Not serious | Not serious | Serious† | None | 288 | 288 | . . . | MD 1.20 higher (0.09 higher to 2.31 higher) | Moderate | Important |
| **Absolute change in DLCO% predicted** | | | | | | | | | | | |
| 2 | Not serious | Serious* | Not serious | Serious† | None | 385 | 398 | . . . | MD 0.54 higher (1.64 lower to 2.71 higher) | Low | Important |
| **Absolute change in 6MWD (m)** | | | | | | | | | | | |
| 2 | Not serious | Not serious | Not serious | Serious† | None | 119 | 128 | . . . | MD 28.69 higher (10.63 higher to 46.75 higher) | Moderate | Important |
| **Absolute change in SGRQ** | | | | | | | | | | | |
| 2 | Not serious | Very serious¶ | Not serious | Serious† | None | 308 | 308 | . . . | MD 1.99 lower (9.14 lower to 5.15 higher) | Very low | Important |
| **Acute exacerbation of ILD** | | | | | | | | | | | |

(*Continued*)

**Table 2.** (Continued)

| No. of Studies | Risk of bias | Inconsistency | Indirectness | Imprecision | Other considerations | Antifibrotic drugs | Placebo | RR (95% CI) | Absolute (95% CI) | Certainty | Importance |
|---|---|---|---|---|---|---|---|---|---|---|---|
| | | **Certainty Assessment** | | | | **No. of Patients** | | **Effect** | | **Certainty** | **Importance** |
| 1 | Not serious | Not serious | Not serious | Serious[†] | None | 4/82 (4.9%) | 8/88 (9.1%) | 0.54 (0.17–1.71) | 41 fewer per 1000 (from 75 fewer to 59 more) | Moderate | Important |
| **AE: diarrhea** | | | | | | | | | | | |
| 3 | Not serious | Not serious | Not serious | Not serious | None | 441/636 (69.3%) | 171/636 (26.9%) | 2.56 (2.23–2.93) | 283 more per 1000 (from 234 more to 332 more) | High | Important |
| **AE: nausea** | | | | | | | | | | | |
| 2 | Not serious | Not serious | Not serious | Not serious | None | 187/620 (30.2%) | 70/619 (11.3%) | 2.65 (2.02–3.49) | 159 more per 1000 (from 102 more to 229 more) | High | Important |
| **AE: vomiting** | | | | | | | | | | | |
| 2 | Not serious | Not serious | Not serious | Not serious | None | 132/620 (21.3%) | 47/619 (7.6%) | 2.81 (1.88–4.20) | 123 more per 1000 (from 62 more to 206 more) | High | Important |
| **AE: elevation of transaminases** | | | | | | | | | | | |
| 2 | Not serious | Very serious[¶] | Not serious | Serious[†] | None | 46/348 (13.2%) | 16/348 (4.6%) | 1.90 (0.44–8.18) | 40 more per 1000 (from 25 fewer to 274 more) | Very low | Important |
| **AEs leading to treatment discontinuation** | | | | | | | | | | | |
| 3 | Not serious | Not serious | Not serious | Not serious | None | 130/747 (17.4%) | 64/743 (8.6%) | 2.02 (1.53–2.68) | 80 more per 1000 (from 43 more to 128 more) | High | Important |
| **Respiratory-related mortality** | | | | | | | | | | | |
| 3 | Not serious | Not serious | Not serious | Serious[†] | None | 4/368 (1.1%) | 7/368 (1.9%) | 0.58 (0.10–3.38) | 8 fewer per 1000 (from 17 fewer to 44 more) | Moderate | Important |

GRADE was based on trials with low risk of bias only.

*The heterogeneity ($I^2$) across studies was of >50%.

[†]TSA suggested uncertain results.

[‡]The $I^2$ across studies was 43% and TSA indicated a diversity $D^2$ of 50%.

[¶]The $I^2$ across studies was of ≥75%.

Abbreviations: DLCO, diffusion capacity of the lung for carbon monoxide; FVC, forced vital capacity; GRADE, Grading of Recommendations, Assessment, Development and Evaluations; ILD, interstitial lung disease; IPF, idiopathic pulmonary fibrosis; MD, mean difference; PF, progressive fibrosis; RCT, randomized controlled trial; RR, risk ratio; SAEs, serious adverse events; SGRQ, St. George's Respiratory Questionnaire; 6MWD, six-minute walk distance.

[10,12,16,17,29]; n = 1650). TSA indicated only 10.9% of the RIS was reached (TSA-adjusted CI 0.06 to 7.14) and the certainty of evidence was moderate.

Consistent results concerning SAEs were indicated in subgroup analyses except for the subgroup taking mycophenolate at baseline (RR 1.71; 95% CI 1.09 to 2.70; $I^2$ = 0%; two trials with low risk; n = 324). TSA for this subgroup presented inconclusive finding, with 45.1% of the RIS accrued (TSA-adjusted CI 0.75 to 3.91).

a)

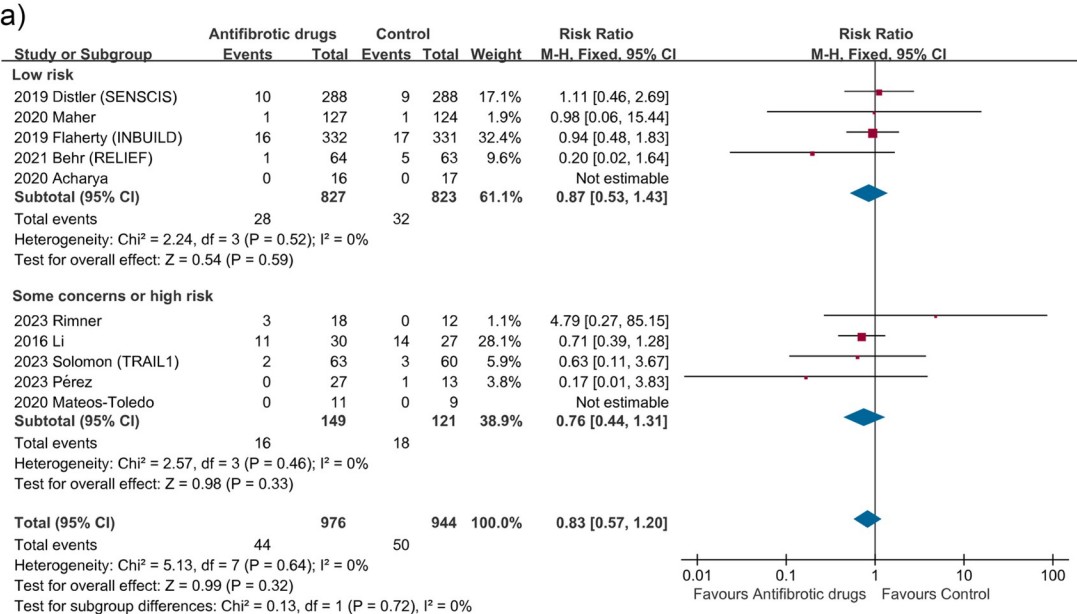

b)

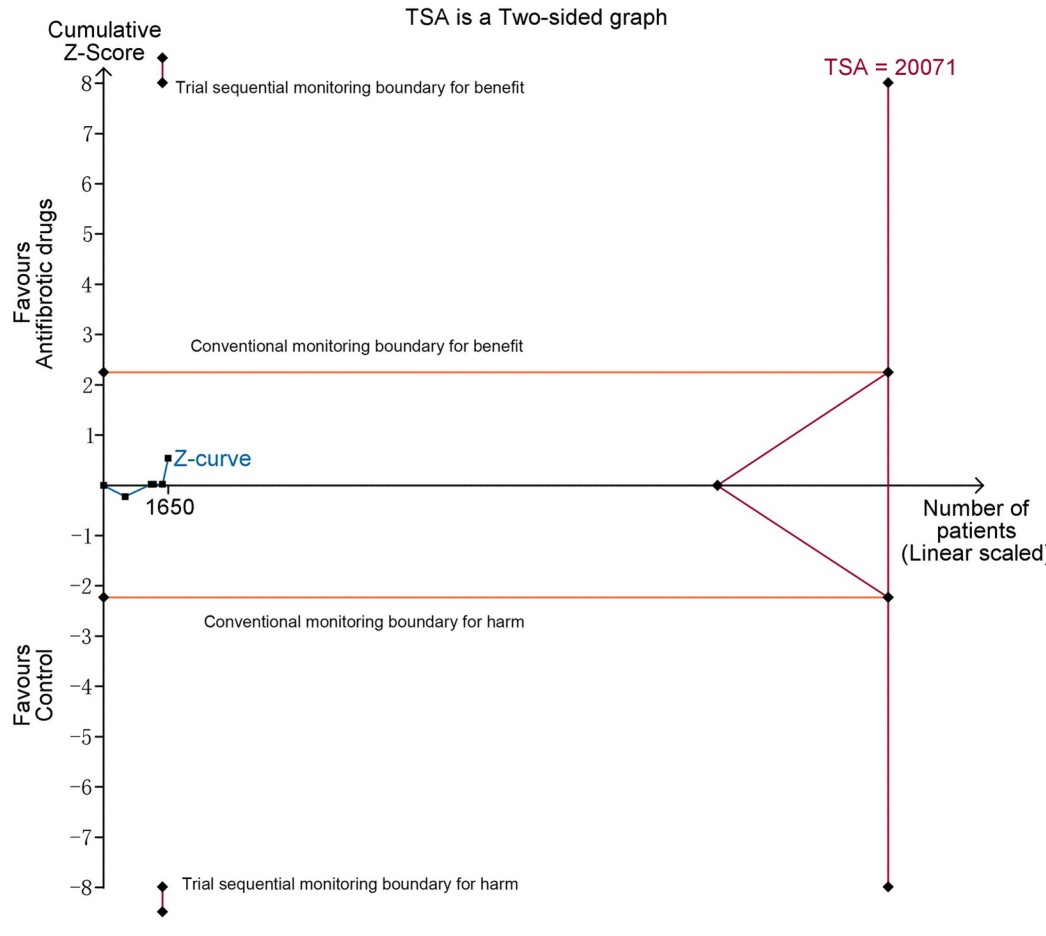

**Fig 3. Meta-analysis and trial sequential analysis (TSA) for all-cause mortality.** (a) Meta-analysis. (b) TSA for trials with low risk of bias. The required information size (RIS) was calculated according to a control event rate of 5.3% (based on all included studies), α of 2.5%, β of 10% and relative risk reduction of 20%. The blue cumulative z curve crossed neither the conventional nor the TSA boundary for benefit, harm or futility, with only 8.2% of the RIS of 20071 patients accrued. Thus the TSA is inconclusive, with TSA-adjusted CI of 0.12 to 6.53 (fixed effect model) and a diversity $D^2$ of 0%.

## Secondary outcomes

A summary of secondary outcomes were presented in Table 2 and S1 File.

**Absolute change in FVC% predicted.** Six studies [15,16,18,30,31,34] reported absolute change in FVC% predicted. Pooled analyses of three trials [15,16,31] with low risk of bias suggested antifibrotic drugs significantly improved absolute decline in FVC% predicted (MD 3.38; 95% CI 1.24 to 5.53; n = 423), consistent with five studies [15,16,18,29,32] on progressive fibrosing ILDs (MD 2.65; 95% CI 1.38 to 3.92; n = 586). TSA highlighted that 50.8% of the RIS was accrued (TSA-adjusted CI 0.34 to 5.86). The certainty of evidence was moderate.

Five studies [10,14,16,18,29] reported absolute decline in FVC ≥ 10% predicted, in which the study of Flaherty et al. reported results with a median follow up of ≥ 12 months [14]. Pooled analyses of four trials [10,14,16,29] with low risk indicated that antifibrotic group had lower risk of absolute decline in FVC ≥ 10% predicted (RR 0.69; 95% CI 0.58 to 0.81; n = 1525) than the control group. TSA showed more than the RIS of 1496 patients was reached (TSA-adjusted CI 0.50 to 0.94). The certainty of evidence was high.

**Annual rate of decline in FVC.** A total of four studies [10,12,18,34] reported annual rates of decline in FVC (ml/yr) and FVC% predicted. Pooled analyses of trials with low risk of bias showed improvements of annual decline rate in FVC (MD 73.39; 95% CI 8.62 to 138.15; two studies on nintedanib [10,12]; n = 1239) and FVC% predicted (MD 1.20; 95% CI 0.09 to 2.31; one study on nintedanib [10]; n = 576) in the antifibrotic group. TSA suggested that 31.2% of the RIS (TSA-adjusted CI -95.04 to 241.81) and 28.5% of the RIS (TSA-adjusted CI -0.75 to 3.15) were accrued for both outcomes respectively. The certainty of evidence was very low for FVC and was moderate for FVC% predicted.

**Acute exacerbation of ILD.** Five studies [15,18,30,34,35] reported acute exacerbation of ILD within 12 months of treatment, which was defined as acute, clinically significant respiratory deteriorations of respiratory symptoms and chest imaging, but not fully explained by cardiac failure or fluid overload. None of these studies indicated significant difference between antifibrotic and control groups, including the only one trial (on nintedanib) [15] with low risk of bias (RR 0.54; 95% CI 0.17 to 1.71; n = 170). TSA indicated 10% of the RIS of 1696 patients was accrued (TSA-adjusted CI 0.00 to 61.56). The certainty of evidence was moderate.

**Other efficacy outcomes.** Pooled analyses of studies with low risk of bias regarding other efficacy outcomes suggested antifibrotic drugs significantly ameliorated the absolute change in 6MWD, but not DLCO% predicted and SGRQ. However, TSA highlighted that the RIS for the three outcomes was not achieved. The certainty of evidence was moderate, low and very low for 6MWD, DLCO% predicted and SGRQ respectively.

**Adverse events.** A total of 10 studies [10,12,16–18,29,30,33–35] reported AEs. Gastrointestinal disorders were the most reported AEs, including diarrhea, nausea and vomiting. Other AEs comprised elevation of transaminases, skin rash and ulcer etc. Pooled analyses of trials with low risk revealed antifibrotic drugs increased risk of diarrhea (RR 2.58; 95% CI 2.25 to 2.95; three studies [10,12,29]; n = 1272), nausea (RR 2.65; 95% CI 2.02 to 3.49; two studies [10,12]; n = 1239) and vomiting (RR 2.81; 95% CI 1.88 to 4.20; two studies [10,12]; n = 1239), but not elevation of transaminases (RR 1.90; 95% CI 0.44 to 8.18; two studies [12,29]; n = 696). TSA showed that more than the RIS was accrued for outcomes of diarrhea, nausea and

a)

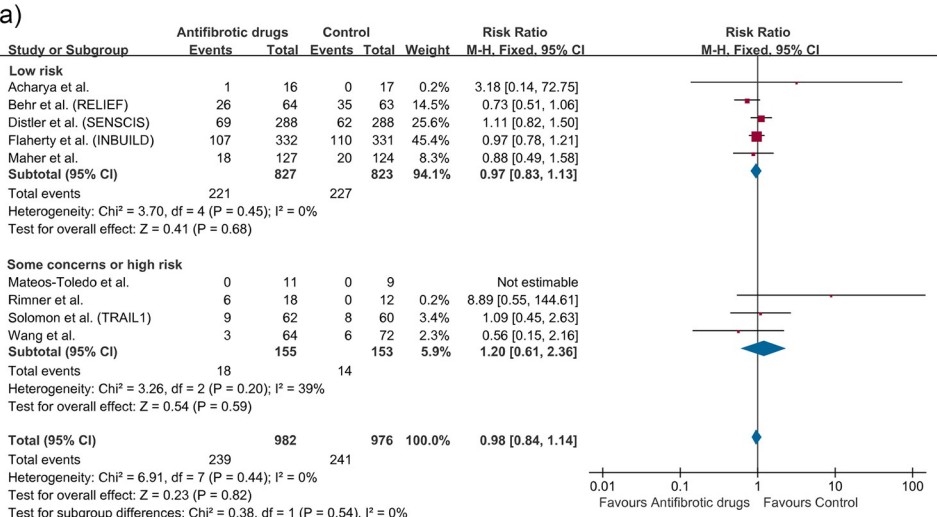

b)

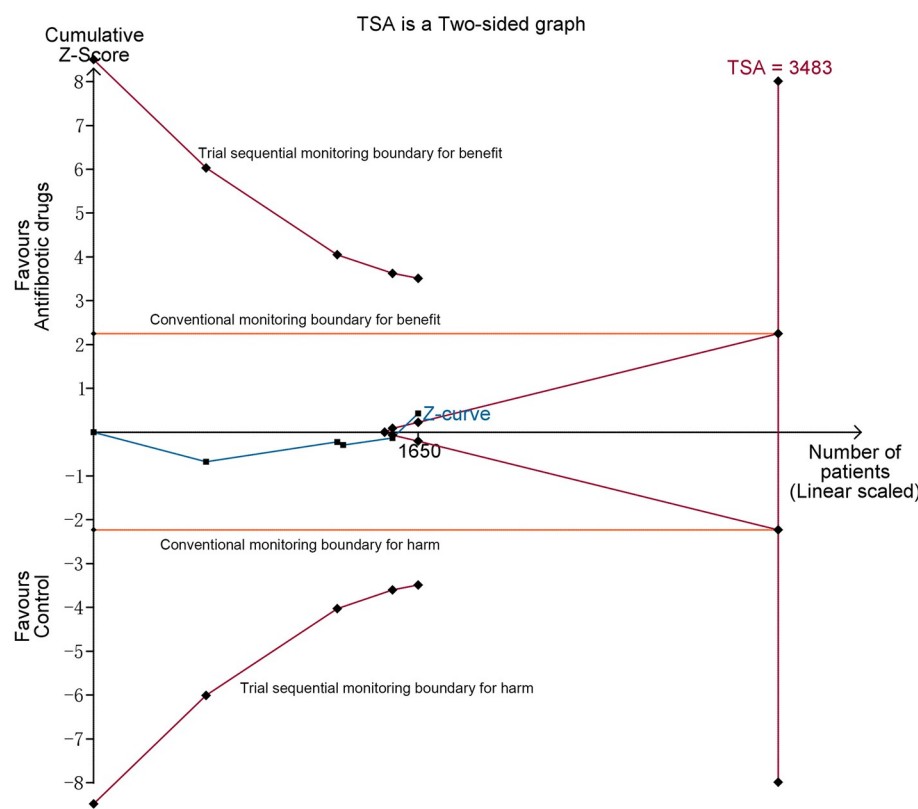

**Fig 4. Meta-analysis and trial sequential analysis (TSA) for serious adverse events.** (a) Meta-analysis. (b) TSA for trials with low risk of bias. The required information size (RIS) was calculated according to a control event rate of 24.92% (based on all included studies), α of 2.5%, β of 10% and relative risk reduction of 20%. The blue cumulative z curve crossed neither the conventional nor the TSA boundary for benefit, harm or futility, with 47.4% of the RIS of 3483 patients accrued. Thus the TSA is inconclusive, with TSA-adjusted CI of 0.74 to 1.28 (fixed effect model) and a diversity $D^2$ of 0%.

vomiting, with high certainty of evidence. However, only 4.3% of the RIS was accrued for transaminases elevating and the certainty of evidence was very low due to substantial heterogeneity and imprecision.

Seven studies [10,12,16,18,30,34,35] reported AEs leading to treatment discontinuation. Pooled analyses of trials with low risk revealed higher risk of AEs leading to discontinuation in antifibrotic group, with a RR of 2.02 compared to control group (95% CI 1.53 to 2.68; three studies [10,12,16]; n = 1490). TSA showed more than the RIS was accrued (TSA-adjusted CI 1.21 to 3.37) and the certainty of evidence was high.

### Exploratory outcome

Eight studies [10,12,17,18,30,33–35] reported respiratory-related death and pooled analyses of three trials [10,17,29] with low risk of bias suggested that antifibrotic drugs were not associated with respiratory-related mortality (RR 0.58; 95% CI 0.10 to 3.38; n = 736). TSA highlighted only 4.6% of the RIS (16152 patients) was accrued. The certainty of evidence was moderate (Table 2).

### Subgroup and sensitivity analyses

Subgroup and sensitivity analyses were presented in the S1 File. Subgroup analyses were almost consistent with the primary analyses, both in all participants and those with progressive fibrosis. Tests for subgroup interaction revealed heterogeneity across different antifibrotic drugs, duration of follow up, ILD subtypes, HRCT patterns, risk of bias of study, between taking and not taking mycophenolate at baseline. Sensitivity analyses suggested robust results except for secondary outcomes including acute exacerbation of ILD (only one study with low bias), change in SGRQ and elevation of transaminases (both with substantial heterogeneity).

## Discussion

In this systematic review and meta-analysis, 17 studies concerning pirfenidone and nintedanib were included. Of them, 11 studies were rated as low risk of bias [10–17,29,31,32]. For the primary outcomes, we found antifibrotic drugs ameliorated absolute decline in FVC after 6 to 12 months of treatment among patients with non-IPF ILDs, including those with a progressive fibrosing phenotype, with moderate certainty, whereas no effect was observed in all-cause mortality and SAEs. However, TSA suggested the RIS for all-cause mortality and SAEs was far from reached. Besides, antifibrotic drugs were also associated with improved FVC% predicted and 6MWD, but not DLCO% predicted, SGRQ, acute exacerbation of ILD and respiratory-related mortality, and had higher risk of gastrointestinal AEs and treatment discontinuation. In subgroup analyses, the benefits of antifibrotic drugs in FVC was not shown in patients taking mycophenolate and this subgroup even had higher risk of SAEs.

The similarity in pathogenesis and clinical progression between IPF and other subtypes makes it possible for the use of nintedanib and pirfenidone in ILDs other than IPF. Currently, clinical trials concerning antifibrotic drugs treating non-IPF ILDs are limited and most with small sample sizes. Possible explanations are the challenging diagnosis [3,36], slow recruitment [17,18] and poor adherence of patients [29] resulting from apparent heterogeneity, rapid progression, high morbidity and mortality of non-IPF ILDs, making it difficult for clinical trials to accrue evidence. In the present study, TSA suggests the RIS for mortality, SAEs, DLCO% predicted and SGRQ is far insufficient. Therefore, in addition to clinical trials, more considerations could be given to the real-world data, which is also essential source of information, particularly for uncommon subtypes such as hypersensitivity pneumonitis and unclassifiable ILD.

Based on the control group across included studies, we found a mortality of up to 51.9% within 6 to 12 months in non-IPF subtypes, comparable to IPF [3,37]. The role of antifibrotic drugs on mortality in IPF has been controversial [38,39], and we also could not draw a firm conclusion in non-IPF ILDs, whether on all-cause mortality or respiratory-related mortality. Of note, unlike IPF, patients with non-IPF ILDs frequently suffer from disorders involving multiple organ-systems, thus all-cause mortality may be less accurate than respiratory-related mortality in the evaluation of antifibrotic therapy. Although the PROGRESS study [9] indicated a significant association between all-cause mortality and FVC decline $\geq$10% predicted (which was correlated with antifibrotic drugs), our knowledge concerning this relationship was still limited. According to the results of TSA, more large-sample studies are needed to verify the effect of antifibrotic therapy on mortality.

The lung function, particularly FVC, has been regarded as the primary indicator of ILD progression and prognosis [40,41]. Based on the INBUILD and SENSCIS trials, both the ATS/ERS/JRS/ALAT and ATS guidelines suggested using nintedanib to treat patients with progressive fibrosis and SSc-ILD, though with low and very low certainty [5,7]. By adding several newly published RCTs [16,31], our findings regarding FVC and FVC% predicted made further validations, with higher certainty of evidence. Although only four studies with low risk of bias were included in the primary analyses, the TSA indicated that these data were sufficient to reach a firm conclusion. Besides, marked heterogeneity across studies was observed when analyzing absolute change in FVC and annual rate of decline in FVC. Further subgroup analyses suggested that antifibrotic drugs, follow-up time, ILD subtypes, HRCT patterns and mycophenolate treatment were possible causes of heterogeneity. However, in these subgroups, the positive association between FVC improvement and antifibrotic therapy did not substantially changed, except for the subgroup taking mycophenolate at baseline. The advantage of antifibrotic drugs in FVC is particularly important for progressive fibrotic ILDs. And in patients with a progressive fibrosing phenotype, consistently significant results were also indicated, with no heterogeneity. Therefore, antifibrotic drugs could be effective in improving FVC for non-IPF ILDs, including AID-ILDs, hypersensitivity pneumonitis and unclassifiable ILD, whereas for ILD subtypes not analyzed in this study and patients taking mycophenolate, their effects remain to be determined.

Another important finding of this study was that patients taking mycophenolate at baseline did not benefit from antifibrotic drugs in FVC and FVC% predicted, but with higher risk of SAEs. As a common immunomodulator, mycophenolate has been approved treating autoimmune diseases and reported beneficial for ILDs other than IPF [7,42,43]. In clinical practice, the use of mycophenolate or combination of mycophenolate plus antifibrotic drugs sometimes are inevitable in AID-ILDs. The ATS guideline also suggests the combination of mycophenolate and nintedanib in patients with SSc-ILD, but with very low certainty [7]. We speculate that the reduced benefit of antifibrotic drugs in patients taking mycophenolate may correlate to ILD subtypes with varied aetiologies and different baseline FVC. In the two studies included for analysis of patients taking mycophenolate, the first study [11] included patients with SSc-ILD, which could benefit from mycophenolate, and the second study [32] investigated unclassifiable ILD, in which the role of mycophenolate was uncertain [44]. Different responses to mycophenolate may contribute to varied baseline FVC and FVC change, as well as AEs. Actually, the mean percentage predicted FVC at baseline in the second study was different between patients taking and not taking mycophenolate [32]. The overlap in AEs such as gastrointestinal events and elevation of transaminases between mycophenolate and antifibrotic drugs may further aggravate the severity of AEs. The higher risk of SAEs across patients taking mycophenolate in our study indicates that the combination of both drugs should be treated with caution

[6,45]. More investigations are needed to determine the timing and combination strategy of the both drugs.

Both 6MWD and SGRQ are associated with the prognosis of ILD. Our pooled analyses suggested ameliorated 6MWD in antifibrotic group, but also with limited data. Similar finding was revealed for another relevant outcome, acute exacerbation of ILD. Acute exacerbation is considered contributing to ILD progression, marked disability and mortality. Until now, the aetiology and development of acute exacerbation are obscure, and effective prevention is also sparse [46]. Clinical trials and real-world data have indicated protective effects of nintedanib and pirfenidone against acute exacerbation in IPF [39,47,48]. Chronic diseases may share similar process of acute exacerbation [3,49]. Nonetheless, the concurrent acute exacerbation of non-IPF ILDs and the primary diseases (such as SSc and rheumatoid arthritis) differs from that in IPF. Efficacy evaluation of antifibrotic therapy for acute exacerbation of IPF may not be fully applied to other subtypes, in which the controlling of primary diseases should be taken into consideration. In our study, evidence regarding acute exacerbation of ILD was sparse because only one RCT with low risk of bias was included for primary analysis, whereas pooled estimates based on studies with some concerns or high risk were less robust.

The statistically insignificant results of SAEs make it possible for the use of antifibrotic drugs in non-IPF ILDs. For SAEs, the cumulative z curve almost close to the area of futility when using a RRR of 20% might imply that risk reduction of 20% in antifibrotic group is unlikely. Additionally, we found that gastrointestinal AEs remained an issue for antifibrotic drugs, particularly for nintedanib. The increased RRs of nausea and vomiting were similar to previous reports in ILD [50,51]. Due to sparse data, we were unable to identify the main AEs leading to discontinuation, although which appeared to be acute exacerbation and transfer to another facility in IPF [52]. Considering the adverse effect of discontinuation on prognosis, specific efforts are required to figure out the main causes of treatment discontinuation.

This study updated previous meta-analyses [19,20] by adding RCTs, prospective studies and post-hoc analyses of RCTs published in 2022 and beyond, promoting the accumulation of evidence. The quality of studies were evaluated rigorously and all conclusions were made based on studies with low risk of bias. To our knowledge, compared with previous evidence, we included more patients, and performed more subgroup analyses based on critical properties of non-IPF ILDs. Besides, apart from meta-analysis, we conducted the TSA to estimate the RIS and precision of results, also applied the GRADE tool to assess the quality of evidence.

However, this study has several limitations. First, the number, sample size and quality of studies were limited, making it difficult to draw firm conclusions for most outcomes in this study. Second, because of sparse data, we were unable to separately assess pirfenidone and nintedanib in patients with different ILD subtypes or phenotypes, though analyses in patients with a progressive fibrosing phenotype were performed. Third, marked heterogeneity was observed in several outcomes. To investigate cause of heterogeneity and further reduce its impact, we further conducted subgroup analyses and made cautious conclusions. However, other factors that we failed to analyze such as severity of disease, duration of medication and background treatment may also weaken the robustness of results. Therefore, these findings could not be generalized to all subtypes of non-IPF ILDs. Fourth, several included RCTs (judged as some concerns or high risk) were terminated early due to slow recruitment or the COVID-19 pandemic [17,18,34], in which the results were based on imputation of missing data and intention-to-treat analysis. This may lead to an underestimation of the significance of results.

In conclusion, this study suggests that pirfenidone and nintedanib could slow disease progression by ameliorating decline in FVC and FVC% predicted in patients with non-IPF ILDs, but may not correlate to all-cause mortality and acute exacerbation of ILD. For safety, although

both drugs are not well-tolerated, they may not increase the risk of SAEs. However, antifibrotic drugs appear having more harm than good in the subgroup taking mycophenolate, indicating more investigations regarding different subtypes are warranted.

## Supporting information

**S1 Checklist. Preferred Reporting Items for Systematic Reviews and Meta-Analyses (PRISMA) checklist.**
(DOCX)

**S1 Table. Study selection.**
(XLSX)

**S2 Table. Extract data details.**
(XLSX)

**S1 File.**
(DOC)

## Author Contributions

**Conceptualization:** Mei Yang, Yuying Tan, Ting Yang, Dan Xu, Mei Chen, Lei Chen.

**Data curation:** Mei Yang, Yuying Tan, Ting Yang.

**Formal analysis:** Mei Yang, Yuying Tan, Ting Yang, Dan Xu.

**Investigation:** Mei Yang, Yuying Tan, Ting Yang, Dan Xu.

**Methodology:** Mei Yang, Yuying Tan, Ting Yang, Dan Xu, Mei Chen.

**Project administration:** Mei Chen, Lei Chen.

**Software:** Ting Yang, Dan Xu.

**Validation:** Ting Yang, Dan Xu.

**Visualization:** Mei Yang, Yuying Tan, Ting Yang, Dan Xu.

**Writing – original draft:** Mei Yang, Yuying Tan, Ting Yang.

**Writing – review & editing:** Dan Xu, Mei Chen, Lei Chen.

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
