## [Decision Letter · Decision Letter 0]

14 Nov 2024

PONE-D-24-44228Efficacy and safety of antifibrotic drugs for interstitial lung diseases other than IPF: a systematic review, meta-analysis and trial sequential analysisPLOS ONE

Dear Dr. Chen,

Thank you for submitting your manuscript to PLOS ONE. After careful consideration, we feel that it has merit but does not fully meet PLOS ONE’s publication criteria as it currently stands. Therefore, we invite you to submit a revised version of the manuscript that addresses the points raised during the review process.

We look forward to receiving your revised manuscript.

Kind regards,

Yoshiaki Zaizen, MD, PhD

Academic Editor

PLOS ONE

2. In the online submission form you indicate that your data is not available for proprietary reasons and have provided a contact point for accessing this data. Please note that your current contact point is a co-author on this manuscript. According to our Data Policy, the contact point must not be an author on the manuscript and must be an institutional contact, ideally not an individual. Please revise your data statement to a non-author institutional point of contact, such as a data access or ethics committee, and send this to us via return email. Please also include contact information for the third party organization, and please include the full citation of where the data can be found.

4. As required by our policy on Data Availability, please ensure your manuscript or supplementary information includes the following:

Reviewers' comments:

Reviewer's Responses to Questions

**Comments to the Author**

1. Is the manuscript technically sound, and do the data support the conclusions?

Reviewer #1: Yes

Reviewer #2: Yes

2. Has the statistical analysis been performed appropriately and rigorously? 

Reviewer #1: Yes

Reviewer #2: Yes

3. Have the authors made all data underlying the findings in their manuscript fully available?

Reviewer #1: Yes

Reviewer #2: Yes

4. Is the manuscript presented in an intelligible fashion and written in standard English?

Reviewer #1: Yes

Reviewer #2: No

5. Review Comments to the Author

Reviewer #1: Dear Authors,

The article needs minor revision and slight corrections to be considered for publication in the PLOS ONE Journal.

Analysis of efficacy and safety of antifibrotic therapy in different ILDs is very important nowadays when new RCTs are ongoing and new results expected in near future. The article is well-written, and I can recommend its publication after the correction suggested below.

I would like to address the following items that should be corrected – lines:

- 56 … “such as autoimmune disease-related ILD (AID-ILD) and exposure-related ILD” – Please add “among other groups”.

- 58 … “Similar to IPF, non-IPF ILDs may develop progressive fibrosis” – Please correct to “progressive pulmonary fibrosis”.

- 74 … “nintedanib in improving annual decline rate of forced vital capacity (FVC)” – References should be put here (listed later for 1) SENSCIS [27, 22]; and 2) INBUILD [23, 26, 29, 31])

- 93 … “and sarcoidosis etc. [17]” – It looks like this reference # 17 is not applicable here; please clarify

- 101-102 + 298-306 + 415 (…Both 6MWD and SGRQ are clinical measures associated with quality of life) … and health-related quality of life (six minute walk distance [6MWD] and St. George’ s Respiratory Questionnaire [SGRQ]) – Please note that 6MWD test does not belong to the HRQL and it should be presented/listed separately from SGRQ throughout the manuscript.

- 103 … “from baseline to study endpoint. Annual” – This should not be a new sentence from “Annual…”

- 137 … “the Newcastle-Ottawa Scale was adopted, with a total score of 9.” – Reference is missing (e.g. Stang A 2010 or another, per authors’ decision)

- 208 … “Patients with UIP was reported in five studies [12, 23, 24, 29, 31].” – Clarification if UIP was related to IPF need to be provided/explained.

- 329 + 332-335 (+ 431 + 464) … “10 studies [10-12, 22-25, 32-34] reported fatal AEs.”; “no statistical significance for fatal AEs” – AEs that lead to death are classified as SAEs, so this should be reported in another section/paragraph (SAEs).

With many thanks and best regards

Reviewer #2: Thank you for the opportunity to review this paper. The authors correctly identify that use of antifibrotics in non-IPF fibrosing lung disease is an area with minimal research, and correctly suggest the need for further work. The systematic review appears to have been carried out in a coherent manner, and the statistical analysis plan appears sound.

The authors correctly identify that the studies available are very heterogenous and small in number, which limits the conclusions that can be made from the analysis. I feel this limitation is understated in the manuscript, and significantly affects the findings in a number of secondary and exploratory outcomes.

Specific feedback points include:

- The range of non-IPF fibrotic lung disease is extremely wide, and there is marked heterogeneity within the studies quoted. With regard to the FVC analysis, for example, only two studies for each antifibrotic meet inclusion criteria, with varied aetiologies. This should be explored further within the discussion of the manuscript.

- Can the authors explain why all-cause mortality results are not divided based on antifibrotic used, i.e. all-cause mortality results for nintenadib and for pirfenidone separately? It is odd this approach is used for FVC decline but not mortality

- Can the authors elaborate on which studies were at low risk of bias and thus chosen for the section on Annual rate of decline in FVC (line 284), and health-related quality of life (line 300)

- A number of the outcomes require analysis in which only a couple of studies are included, which are significantly heterogenous in terms of population and study design, e.g. Absolute change in DLCO% Predicted, Health-related quality of life. I would suggest cutting these sections of the manuscript or including them within an appendix

- Please state how many studies had low risk of bias within the first paragraph of your discussion, as these were used to form the base of the analysis (page 17)

- Please rewrite lines 363-365, as the syntax does not make sense

- Are the authors able to suggest any hypothesis as to why the patient group taking mycofenolate had reduced benefits and increased risk of harm? Do you feel it is related to a difference in underlying aetiology?

- The manuscript would benefit from some rewriting particularly in terms of syntax and grammar. This may be addressed during the copy-editing process

6. PLOS authors have the option to publish the peer review history of their article (what does this mean?). If published, this will include your full peer review and any attached files.

Reviewer #1: No

Reviewer #2: No

---

## [Author Response · Author response to Decision Letter 0]

14 Dec 2024

Dear editor and reviewers:

It is an honor to receive the revisal invitation. In order to meet your requirements, we have tried our best to reply all the comments very carefully (as follows) and revised the manuscript.

Due to limited space, we present the main results in the manuscript. And all the analyses of each outcome (meta-analysis, TSA, subgroup and sensitivity analyses, risk of bias assessment) are presented in the Appendix in detail.

Reviewers’ comments and responses

Reviewer #1: Dear Authors,

The article needs minor revision and slight corrections to be considered for publication in the PLOS ONE Journal.

Analysis of efficacy and safety of antifibrotic therapy in different ILDs is very important nowadays when new RCTs are ongoing and new results expected in near future. The article is well-written, and I can recommend its publication after the correction suggested below. I would like to address the following items that should be corrected:

1. Line 56 … “such as autoimmune disease-related ILD (AID-ILD) and exposure-related ILD” – Please add “among other groups”.

Response: We have added this phrase to reflect the reviewer comment (Lines 56-57 of the manuscript).

2. Line 58 … “Similar to IPF, non-IPF ILDs may develop progressive fibrosis” – Please correct to “progressive pulmonary fibrosis”.

Response: We have made corresponding correction (Line 58).

3. Line 74 … “nintedanib in improving annual decline rate of forced vital capacity (FVC)” – References should be put here (listed later for 1) SENSCIS [27, 22]; and 2) INBUILD [23, 26, 29, 31])

Response: We have made corresponding corrections (Lines 71-72).

4. Line 93 … “and sarcoidosis etc. [17]” – It looks like this reference # 17 is not applicable here; please clarify.

Response: We have made corresponding correction (Line 93).

5. Lines 101-102 + 298-306 + 415 (…Both 6MWD and SGRQ are clinical measures associated with quality of life) … and health-related quality of life (six minute walk distance [6MWD] and St. George’ s Respiratory Questionnaire [SGRQ]) – Please note that 6MWD test does not belong to the HRQL and it should be presented/listed separately from SGRQ throughout the manuscript.

Response: Thank you for raising this point. We have made corresponding corrections (Lines 101-102). Due to limited space, we present the results of 6MWD, DLCO% predicted and SGRQ in the Appendix in detail and make a brief summary in the main text (Lines 305-309).

6. Line 103 … “from baseline to study endpoint. Annual” – This should not be a new sentence from “Annual…”

Response: We have made corresponding correction (Line 102).

7. Line 137 … “the Newcastle-Ottawa Scale was adopted, with a total score of 9.” – Reference is missing (e.g. Stang A 2010 or another, per authors’ decision)

Response: We have added corresponding reference (Line 137).

8. Line 208 … “Patients with UIP was reported in five studies [12, 23, 24, 29, 31].” – Clarification if UIP was related to IPF need to be provided/explained.

Response: This study focuses on ILDs other than IPF, and patients are diagnosed with non-IPF ILDs as per the included original studies. The diagnosis and key eligibility criteria of patients across included studies are presented in the S5 Appendix. Thus the UIP reported in our study is not related to IPF, which is also clarified in the study title and eligibility criteria (Lines 92-94).

9.Lines 329 + 332-335 (+ 431 + 464) … “10 studies [10-12, 22-25, 32-34] reported fatal AEs.”; “no statistical significance for fatal AEs” – AEs that lead to death are classified as SAEs, so this should be reported in another section/paragraph (SAEs).

Response: Thank you for raising this point. We have made corresponding correction (Lines 260-264).

Reviewer #2: Thank you for the opportunity to review this paper. The authors correctly identify that use of antifibrotics in non-IPF fibrosing lung disease is an area with minimal research, and correctly suggest the need for further work. The systematic review appears to have been carried out in a coherent manner, and the statistical analysis plan appears sound. The authors correctly identify that the studies available are very heterogenous and small in number, which limits the conclusions that can be made from the analysis. I feel this limitation is understated in the manuscript, and significantly affects the findings in a number of secondary and exploratory outcomes. Specific feedback points include:

1. The range of non-IPF fibrotic lung disease is extremely wide, and there is marked heterogeneity within the studies quoted. With regard to the FVC analysis, for example, only two studies for each antifibrotic meet inclusion criteria, with varied aetiologies. This should be explored further within the discussion of the manuscript.

Response: Considering the heterogeneity across studies and limited data, we made the conclusions with caution, which were mentioned in the Discussion section (Line 449-450, 453-455). As shown in the Appendix S2 Table, marked heterogeneity (≥ 50%) was observed for the outcomes of change in FVC, DLCO% predicted and SGRQ, elevation of transaminases. Further subgroup analyses were performed to explain causes of heterogeneity and test robustness of results. For the FVC analysis, results of subgroup analyses were consistently significant except for the subgroup taking mycophenolate at baseline, and most importantly, the results of FVC analysis were also consistent with FVC% predicted. Besides, although limited studies included for FVC analysis, the TSA suggested the required sample size was accrued. Therefore, the conclusion regarding the primary outcome, change in FVC, is based on objective results. For other outcomes with apparent heterogeneity, combined with TSA, we have highlighted the need for further investigations in the Discussion section.

We acknowledge that the aetiology of ILD could be one of the main causes of heterogeneity and corresponding subgroup analyses are also conducted (S2-S3 Tables). Due to the diverse and complex nature of ILDs, it is necessary to first analyze their commonalities. As stated in the Introduction section (Lines 57-59), although with different aetiologies, inflammation and fibrosis are considered as the primary pathophysiological features of ILDs, which are the therapeutic targets for both antifibrotic drugs, making our study meaningful. We have added some discussions about heterogeneity and limited evidence to reflect the reviewer comment (Lines 384-397, 418-419, 449-451, 453-458).

2. Can the authors explain why all-cause mortality results are not divided based on antifibrotic used, i.e. all-cause mortality results for nintenadib and for pirfenidone separately? It is odd this approach is used for FVC decline but not mortality

Response: We conducted subgroup analyses regarding antifibrotic drugs for all reported outcomes, including all-cause mortality, and reported results of subgroup analyses in the main text (Lines 336-340). Due to limited space, the tables summarizing all the subgroup analyses were presented in the Appendix S2-S3 Tables.

As shown in S2-S3 Tables, subgroup analyses of antifibrotic drugs revealed that pirfenidone improved all-cause mortality of non-IPF ILDs while nintedanib not. However, as stated in the Result and Discussion sections (Lines 243-245, 374-378), we could not draw a firm conclusion on all-cause mortality because of: 1) insufficient RIS suggested by TSA; 2) the inclusion of studies with some concerns or high risk of bias in the subgroup analyses of antifibrotic drugs; 3) limited accuracy of all-cause mortality (Lines 372-374).

3. Can the authors elaborate on which studies were at low risk of bias and thus chosen for the section on Annual rate of decline in FVC (line 284), and health-related quality of life (line 300)

Response: Due to limited space, studies with low risk bias were presented in the S6-S8 Appendix and S1 Table. We used the Cochrane Collaboration tool, RoB2, to assess the quality of study. As shown below, for annual rate of decline in FVC, there were two studies with low risk of bias. For 6MWD and SGRQ, there were also two studies with low risk respectively, as presented in the Appendix S1.10 and S1.11 Tables.

4.A number of the outcomes require analysis in which only a couple of studies are included, which are significantly heterogenous in terms of population and study design, e.g. Absolute change in DLCO% Predicted, Health-related quality of life. I would suggest cutting these sections of the manuscript or including them within an appendix

Response: We have made corresponding corrections. We presented detailed results of these secondary outcomes in the S7 Appendix and made a brief summary in the main text (Lines 305-309).

5. Please state how many studies had low risk of bias within the first paragraph of your discussion, as these were used to form the base of the analysis (page 17)

Response: We have stated the number of studies with low risk of bias in the Discussion section (Line 347), which are also presented in S1 Table in detail.

6. Please rewrite lines 363-365, as the syntax does not make sense

Response: We have rewritten these lines to reflect the reviewer comment (Line 352-355).

7. Are the authors able to suggest any hypothesis as to why the patient group taking mycofenolate had reduced benefits and increased risk of harm? Do you feel it is related to a difference in underlying aetiology?

Response: We have added some discussions to this finding (Lines 404-417). We speculate that the reduced benefit of antifibrotic drugs in patients taking mycophenolate may correlate to ILD subtypes with varied aetiologies and different baseline FVC. The overlap in adverse reactions such as gastrointestinal events and elevation of transaminases between mycophenolate and antifibrotic drugs may further aggravate the severity of adverse events.

8. The manuscript would benefit from some rewriting particularly in terms of syntax and grammar. This may be addressed during the copy-editing process

Response: Thank you for the suggestion. We have reviewed the manuscript and made corresponding corrections to minimize syntactic and grammatical errors as much as possible.

All the revisions were marked in red in the revised manuscript with track changes.

All authors have checked the revised manuscript and approved to resubmit to the PLOS ONE. Thank you very much for your consideration.

Sincerely yours,

Lei Chen, MD

Department of Pulmonary and Critical Care Medicine

West China Hospital, Sichuan University

Chengdu, Sichuan 610041, P. R. China

---

## [Decision Letter · Decision Letter 1]

23 Jan 2025

Efficacy and safety of antifibrotic drugs for interstitial lung diseases other than IPF: a systematic review, meta-analysis and trial sequential analysis

PONE-D-24-44228R1

Dear Dr. Chen,

We’re pleased to inform you that your manuscript has been judged scientifically suitable for publication and will be formally accepted for publication once it meets all outstanding technical requirements.

Kind regards,

Yoshiaki Zaizen, MD, PhD

Academic Editor

PLOS ONE

Additional Editor Comments (optional):

Congratulations and we thank you for your valuable contribution to PLOS One.

Reviewers' comments:

Reviewer's Responses to Questions

**Comments to the Author**

1. If the authors have adequately addressed your comments raised in a previous round of review and you feel that this manuscript is now acceptable for publication, you may indicate that here to bypass the “Comments to the Author” section, enter your conflict of interest statement in the “Confidential to Editor” section, and submit your "Accept" recommendation.

Reviewer #1: All comments have been addressed

Reviewer #2: All comments have been addressed

2. Is the manuscript technically sound, and do the data support the conclusions?

Reviewer #1: Yes

Reviewer #2: Yes

3. Has the statistical analysis been performed appropriately and rigorously? 

Reviewer #1: Yes

Reviewer #2: Yes

4. Have the authors made all data underlying the findings in their manuscript fully available?

Reviewer #1: Yes

Reviewer #2: Yes

5. Is the manuscript presented in an intelligible fashion and written in standard English?

Reviewer #1: Yes

Reviewer #2: Yes

6. Review Comments to the Author

Reviewer #1: Dear Authors,

I am happy to see that ALL my comments and questions are properly addressed. I do not have any additional requests to the authors. Therefore, I recommend that this manuscript can proceed in the submission process, i.e. it can be accepted for publication as it appears now.

With many thanks and best regards

Reviewer #2: Thank you for addressing the suggestions contained within my review.

Best of luck with your ongoing work in this field.

7. PLOS authors have the option to publish the peer review history of their article (what does this mean?). If published, this will include your full peer review and any attached files.

Reviewer #1: No

Reviewer #2: No

---

## [Editor Report · Acceptance letter]

30 Jan 2025

PONE-D-24-44228R1 

PLOS ONE

Dear Dr. Chen, 

I'm pleased to inform you that your manuscript has been deemed suitable for publication in PLOS ONE. Congratulations! Your manuscript is now being handed over to our production team.

Kind regards, 

on behalf of

Dr. Yoshiaki Zaizen 

Academic Editor

PLOS ONE